# Engineering of Humanized PSMA-Directed CAR T Cells for Potent and Specific Elimination of Prostate Cancer Cells

**DOI:** 10.3390/cells14171333

**Published:** 2025-08-28

**Authors:** Tobias D. Deller, Jamal Alzubi, Laura Mosti, Marie Peschers, Christian Gratzke, Philipp Wolf, Toni Cathomen

**Affiliations:** 1Institute for Transfusion Medicine and Gene Therapy, Medical Center-University of Freiburg, 79106 Freiburg, Germanyjamal.alzubi@uniklinik-freiburg.de (J.A.); mosti.laura@gmail.com (L.M.); 2Center for Cell and Gene Therapy, Medical Center-University of Freiburg, 79106 Freiburg, Germany; 3Faculty of Medicine, University of Freiburg, 79106 Freiburg, Germany; 4Department of Haematooncology, Faculty of Medicine, University of Ostrava, 70852 Ostrava, Czech Republic; 5Department of Urology, Medical Center-University of Freiburg, 79106 Freiburg, Germany; 6Faculty of Biology, University of Freiburg, 79104 Freiburg, Germany

**Keywords:** CAR T cells, immunogenicity, humanization, prostate cancer, prostate-specific membrane antigen, T cell activation, T cell differentiation, T cell exhaustion

## Abstract

Chimeric Antigen Receptor (CAR) T cell therapy has achieved high response rates in patients with relapsed or refractory hematologic malignancies. However, comparable efficacy in solid tumors remains limited, partly due to poor CAR T cell persistence and immune-mediated rejection. A major contributor, which has hampered the clinical efficacy of CAR T cells in clinical practice, is the immunogenicity of the murine-derived single-chain variable fragments (scFvs) commonly used in CAR constructs. Cell and humoral immune responses to the murine parts of CARs have been implicated in CAR T cell rejection. Here, we describe the generation and in vitro characterization of humanized CAR T cells targeting prostate-specific membrane antigen (PSMA) on prostate cancer cells, based on two distinct murine scFvs (A5 and D7). Humanization improved the germinality index and successfully preserved CAR surface expression. Functional assays demonstrated that humanized PSMA-CAR T cells retained antigen-specific binding, activation and cytotoxicity, differentiation, exhaustion and cytokine secretion profiles comparable to their murine counterparts. These results support the feasibility of humanization as a strategy to reduce immunogenicity without compromising CAR T cell capabilities, providing a foundation for further in vivo validation in solid tumor settings.

## 1. Introduction

Chimeric Antigen Receptor (CAR) T cells are approved for the treatment of hematological malignancies and show great promise. CAR T cell persistence has been shown to correlate with therapy outcomes: higher persistence is associated with greater expansion, fewer relapses, and longer progression-free survival [1,2]. In contrast, solid tumor-targeting CAR T cells have demonstrated limited efficacy. To achieve therapeutic relevance in this setting, strategies to improve CAR T cell persistence [1,2] in the immunosuppressive tumor microenvironment (TME) are needed [3,4,5].

The CAR antigen-binding moiety typically consists of a single-chain variable fragment (scFv) derived from murine antibodies [6]. While this scFv secures target selection, it introduces potentially immunogenic sequences that could trigger a host immune response. Human-anti-mouse-antibodies (HAMAs) can arise after previous exposure to therapeutic antibodies or during CAR T cell treatment, leading to CAR rejection [7,8]. Such immune responses (host-versus-graft) have been detrimental in clinical trials. In a phase I study targeting TAG-72, a humoral anti-CAR response abrogated CAR T cell function [9], and, in another clinical trial targeting ovarian cancer, CAR T cell numbers declined rapidly post-infusion [10]. Severe reactions, including anaphylaxis, have also been reported [11]. Cellular immune responses against murine scFv regions or vector components have similarly compromised CAR T cell efficacy [12,13,14]. Repeated dosing further amplifies these responses, limiting therapeutic impact [13,15]. Humanization of murine scFvs, for example, by grafting complementarity-determining regions (CDRs) onto human germline frameworks, reduces immunogenicity while maintaining antigen specificity [16,17]. Clinical studies have shown that humanized CARs can improve persistence, even following failure of murine CAR therapies [18,19]. However, even minor structural changes may alter CAR behavior, including its surface expression and signaling, or persistence of the resulting CAR T cells [20,21,22,23,24]. These effects are difficult to predict, emphasizing the need for functional validation of each construct.

In this study, we generated humanized versions of two distinct murine PSMA-targeting CARs (scFvs: A5 and D7) [25,26] and characterized their function on different prostate cancer cell lines. We demonstrate that the humanized CAR T cells maintain antigen-specific activation and cytotoxicity, differentiation, exhaustion, and cytokine secretion profiles comparable to their murine counterparts.

## 2. Materials and Methods

### 2.1. Culturing of Target Prostate Cancer Cell Lines

Primary T cells were cultured as previously described [25]. The PSMA-positive cell lines C4-2 (ATCC, CRL-3314, Manassas, VA, USA), LNCaP (ATCC, CRL-1740) and PC3-PSMA (kindly provided by Prof. Paloma H. Giangrande, University of Iowa) [27], and the PSMA-negative cell line DU145 (DSMZ, ACC-261, Braunschweig, Germany) were maintained in RPMI 1640 medium (Gibco, Thermo Fisher Scientific, Waltham, MA, USA) with supplemental penicillin (100 U/mL), streptomycin (100 mg/L) (Sigma-Aldrich, Munich, Germany) and 10% fetal calf serum (FCS) (PAN-Biotech GmbH, Aidenbach, Germany). All cells were cultured in a humidified atmosphere of 5% CO_2_ at 37 °C.

### 2.2. CAR Design and Preparation of γ-Retroviral Vectors

Humanized scFvs of the previously described A5 and D7 scFv sequences were generated by YUMAB GmbH (Braunschweig, Germany) [26,28]. Humanization was achieved by grafting the murine CDR sequences into appropriate human acceptor frameworks. Lead candidates were chosen based on the germinality indexes (GIs) of the humanized light (VL) and heavy (VH) chains (total GI was calculated as the mean of VL and VH GIs) as well as their antigen-specific binding properties. Genes of the murine and humanized VH and VL chains were synthesized, cloned into expression vectors containing constant domains of a human IgG1 antibody for whole heavy and light chain expression in mammalian cells, respectively. Purified antibodies were provided by YUMAB GmbH for further testing.

Specific binding of the antibodies to C4-2 and DU145 cells was evaluated by flow cytometry, as previously described [29]. An hIgG1 isotype antibody (Thermo Fisher Scientific) was used as control. Viable cells were stained with eBioscience™ Fixable Viability Dye eFluor™ 450 (Thermo Fisher Scientific). For antibody detection, a goat anti-human Ig (H + L) antibody conjugated to R-phycoerythrin (RPE) was used (Southern Biotech, Birmingham, AL, USA). Mean fluorescence intensity (MFI) values of stained cells were acquired using a FACSymphony A1 flow cytometer and analyzed with FlowJo software (BD Biosciences, Heidelberg, Germany). Binding affinities were determined using GraphPad Prism 7 software (GraphPad Software Inc., San Diego, CA, USA), with dissociation constants (K_D_) defined as the antibody concentrations that resulted in half-maximal binding.

For CAR T cell production, the VH/VL combinations were cloned into the backbone of our previously tested second-generation CAR [25]. Expression was controlled by an EFS promoter. Additional CAR sequence optimizations included an optimized IgG1 hinge domain and a modified Lck binding moiety in the intracellular CD28 signaling domain [30,31]. Retroviral vectors were generated as previously described [32]. Titers were determined through a serial transduction of Jurkat cells, followed by anti-human IgG staining for CAR expression.

### 2.3. Generation of CAR T Cells

Peripheral blood mononuclear cells (PBMCs) were isolated from leukocyte reduction system (LRS) chambers obtained after informed consent from the blood donation center of the Medical Center-University of Freiburg, and used for the generation of primary CAR T cells as previously described, with modifications [25]. To summarize, thawed PBMCs were kept in RPMI complete medium supplemented with 100 U/mL of IL-2 (ImmunoTools GmbH, Friesoythe, Germany), activated with anti-CD2/CD3/CD28 antibodies (ImmunoCult, STEMCELL Technologies Inc., Vancouver, BC, Canada), and cultured for 3 days before transduction. To improve transduction efficiency, wells were coated with Poly-D-Lysine (PDL, Sigma Aldrich) overnight. For transduction of T cells between 100 and 300 transducing units/cell was used. The medium used for transduction contained 5 µg/mL of protamine sulfate (Sigma Aldrich) and 1000 U/mL of IL-2 (ImmunoTools GmbH). Medium was changed for the first time on day 1 after transduction. The cells were expanded in RPMI complete medium supplemented with 100 U/mL IL-2 (ImmunoTools GmbH) for 10–15 days before being used in experiments and frozen in liquid nitrogen until further use.

### 2.4. Phenotyping of CAR T Cells

CAR surface expression was determined through flow cytometry. Transduced cells were stained for CAR and CD3 expression using anti-human IgG-phycoerythrin (PE) (Southern Biotech) and CD3-allophycocyanin (APC) (Miltenyi Biotec, Bergisch Gladbach, Germany). (CAR) T cells were assessed for different phenotypes after co-culturing with PSMA-positive and PSMA-negative target cells in a 1:1 effector to target ratio for 24 h (differentiation and exhaustion assays) or 48 h (activation assay). Activation of CAR T cells was determined by measuring the expression of CD25 using flow cytometry (CD25-APC, BD Biosciences). T cell differentiation status was determined through expression of CD45RA and CD62L using flow cytometry (anti-human CD62L-Brilliant Violet 421 (BD Biosciences), anti-human CD45RA-fluorescein isothiocyanate (FITC) (BioLegend, San Diego, CA, USA)). T cell exhaustion was determined through measurement of CD223 and CD279 (anti-human CD279-FITC (BD Biosciences), anti-human CD223-eFluor 710 (BD Biosciences)) expression levels.

### 2.5. Cytotoxicity of CAR T Cells

To assess the cytotoxicity of the CAR T cells, they were co-cultured with PSMA-positive and PSMA-negative target cells for 48 h in effector to target ratios from 1:1 to 1:64. After 48 h the viability of the remaining cells was assessed by XTT assays as described previously [33]. To summarize, colorimetric changes mediated by viable cells were quantified using an ELISA reader (Infinite F50, Tecan, Männedorf, Switzerland) at 450 nm. Cytotoxicity in percent was calculated as Equation (1), with OD as the optical density.(1)$$Cytotoxicity\;[\%]=100-\left(\frac{{OD}_{Effector\;+\;Target\;Co\;\text{-}\;Culture}\;-\;{OD}_{Effector\;only}}{{OD}_{Target\;only}\;-\;{OD}_{Medium\;only}}\times 100\right)$$

### 2.6. Cytokine Release Profile of CAR T Cells

(CAR) T cells were co-cultured with PSMA-positive target cells in a 1:1 effector to target ratio for 48 h. Supernatants were collected, and secreted cytokines (TNF, IL-2, IFNγ, GM-CSF, and Granzyme B) were measured by a cytometric bead assay (CBA, BD Biosciences). The CBA was performed according to the manufacturer’s instructions.

### 2.7. Statistical Analysis

The statistical analysis was performed using GraphPad Prism 7 software (GraphPad Software Inc.). The statistical tests used are denoted in the figure captions.

## 3. Results

### 3.1. Humanization of CAR Sequences

Increasing CAR T cell persistence is a key strategy to enhance therapeutic efficacy [34]. To reduce the immunogenicity of our PSMA-targeting CARs, an in silico CDR grafting approach was employed to humanize the two murine scFvs mA5 and mD7. Lead humanized candidates—hA5 and hD7—were selected based on preserved and specific PSMA-binding capacity and increased germinality indexes (GIs). The GIs, reflecting variable heavy chain (VH) and variable light chain (VL) sequence identities to closest human germline genes, improved from 68% (mD7 VH) to 100% (hD7 VH) and from 85% (mD7 VL) to 100% (hD7 VL), resulting in total GIs from 77% for the murine D7 VH/VL combination to 100% for the humanized D7 VH/VL combination. For A5, the GIs increased from 79% (mA5 VH) to 93% (hA5 VH) and from 72% (mA5 VL) to 90% (hA5 VL), resulting in an improved total GI from 76% for the murine A5 VH/VL combination to 92% for the humanized A5 VH/VL combination (Figure 1A).

In general, the respective full antibodies with the murine or humanized D7 VH/VL combinations exhibited higher binding affinities to cellular PSMA compared to the antibodies with the murine or humanized A5 chains. Humanization, however, changed the binding affinities considerably. The humanized antibody with the hD7 VH/VL combination showed approximately a 6-fold reduced binding affinity to C4-2 cells than the murine variant (K_D_ = 1.05 µg/mL vs. K_D_ = 0.17 µg/mL). In contrast, humanization of the A5 variant led to a more than 47-fold increased binding affinity (K_D_ = 1.05 µg/mL vs. K_D_ > 48.9 µg/mL) (Figure 1B). Importantly, both humanized antibodies did not show any binding to antigen-negative cells (Appendix A), indicating the development of specific humanized PSMA-targeting antibodies.

The humanized VH/VL combinations were successfully integrated as scFvs into a second-generation CAR backbone under the elongation factor 1α short (EFS) promoter (Figure 1C). The CAR constructs included a modified IgG1 hinge (minimizing Fc-receptor binding), a CD28 transmembrane and costimulatory domain (deficient in Lck binding), and CD3ζ signaling (Figure 1C) [25,30,31].

All constructs were functionally expressed following γ-retroviral transduction of activated T cells. CAR surface expression was confirmed, with mD7 showing the highest fraction of CAR-positive T cells as well as the highest expression levels (CAR-MFI, representative example, Figure 1D: mD7: 19915, hD7: 9354, mA5: 8471, and hA5: 8166). The other three constructs exhibited lower but comparable expressions (Figure 1D).

### 3.2. CAR T Cell Activation

To evaluate antigen-specific activation, CAR T cells were co-cultured with PSMA-positive C4-2 and PSMA-negative DU145 cell lines. CD25 expression was measured as a surrogate activation marker (Figure 2). 

Following co-culture with PSMA-positive C4-2 cells, all CAR T cells showed a robust activation as indicated by the upregulation of CD25 expression in more than 90% of cells. In contrast, untreated T cells (UT) showed minimal activation (~8%). This activation profile was highly dependent upon the presence of PSMA. All CAR T cell constructs showed significantly higher percentages of CD25-positive cells, as well as significantly higher CD25 expression levels when co-cultured with C4-2 cells as compared to a co-culture with PSMA-negative DU145 cells (Figure 2A,B). When co-cultured with DU145 cells, CD25 levels remained low across all conditions. The hA5 CAR T cells showed CD25 expression levels closest to UT cells, while mA5 and both D7 constructs exhibited slightly elevated CD25 expression (Figure 2A,B). These results indicate that both murine and humanized PSMA-targeting CAR T cells are specifically activated by cognate antigen exposure with minimal off-target activation in PSMA-negative settings and low tonic signaling.

### 3.3. CAR T Cell Cytotoxicity

To assess antigen-specific killing, CAR T cells were co-cultured with PSMA-positive (C4-2, LNCaP, PC3-PSMA) and PSMA-negative (DU145) prostate cancer cell lines across a range of effector-to-target ratios (E:T; 1:1 to 1:64). Cytotoxicity was measured using the XTT assay (Figure 3).

All generated CAR T cells mediated high cytotoxicity against PSMA-positive C4-2 target cells at a 1:1 E:T ratio, with 90% (mD7), 87% (hD7), 94% (mA5), and 89% (hA5), respectively, of C4-2 cells being lysed. At lower E:T ratios, performance differences became apparent. The D7-based CAR T cells, particularly mD7 CAR T cells, maintained higher killing efficacy compared to A5-based CAR T cells. For example, at a 1:4 E:T ratio, mD7 achieved 61% lysis versus 14% for mA5 CAR T cells. When co-cultured with C4-2 cells, mD7 CAR T cells mediated similar cytotoxicity as hD7 CAR T cells (*p* = 0.999), as did mA5 CAR T cells compared to hA5 CAR T cells (*p* = 0.0911). In co-cultures with LNCaP, PC3-PSMA, and DU145, the D7 or A5-derived murine and humanized CAR T cells also showed comparable cytotoxicity profiles (LNCaP: mD7 vs. hD7 *p* = 0.9987, mA5 vs. hA5: *p* = 0.3316; PC3-PSMA: mD7 vs. hD7 *p* = 0.6312, mA5 vs. h *p* = 0.1585; and DU145: mD7 vs. hD7 *p* = 0.9747, mA5 vs. h *p* = 0.9334). Compared to C4-2 and LNCaP cells, the PC3-PSMA target cell line was efficiently lysed already at lower E:T ratios, consistent with its higher PSMA expression levels (Figure 3C, Appendix A). No significant cytotoxicity of CAR T cells was observed against the PSMA-negative DU145 cells (Figure 3D), and untreated T cells did not reveal overt cytotoxicity against any of the cell lines. In conclusion, our results confirm that both murine and humanized PSMA-CAR T cells mediate potent, antigen-specific cytotoxicity in vitro, with differences in efficacy becoming apparent under limiting conditions. Overall, humanization of the CAR scFvs did not impair antigen-specific cytotoxicity. No significant differences between either the mD7 and hD7 CAR T cells or between the mA5 and the hA5 CAR T cells were observed.

### 3.4. CAR T Cell Differentiation

Differentiation status was assessed after co-culture of the CAR T cells with PSMA-positive target cells by evaluating CD45RA and CD62L expression to define naïve/stem cell memory-like (Tn, scm), central memory (Tcm), effector memory (Tem), and effector-like (Teff) subsets (Figure 4).

Compared to UT cells, all CAR constructs exhibited substantial differentiation with increased proportions of effector-like cells. The D7 CAR T cells showed a pronounced differentiation, retaining less than 16% Tn, scm-like cells. In contrast, A5-based CAR T cells maintained a higher proportion of less differentiated Tn, scm-like cells: mA5 (27%) and hA5 (26%). UT cells showed a distinct profile, with a minimal Teff (6%) population. In summary, our humanized CAR T cells preserve a substantial proportion of naïve and memory T cell subsets upon antigen sensitization, which might be favorable to enhance persistence of CAR T cells and therapeutic index [35].

### 3.5. CAR T Cell Exhaustion

To evaluate CAR T cell exhaustion, expression levels of the inhibitory receptors PD-1 (CD279) and LAG-3 (CD223) were measured following co-culture with PSMA-positive target cells (Figure 5). All CAR T cell constructs exhibited increased expression of PD-1 and LAG-3 compared to UT cells, of which 65% were double-negative (Figure 5A). Among the constructs, mD7 CAR T cells showed the highest proportion of double-positive cells (69%) and the lowest proportion of double-negative cells (3%), indicating a more exhausted phenotype. In contrast, hD7, mA5, and hA5 retained higher frequencies of PD-1^−^/LAG-3^−^ cells (10%, 11%, and 8%, respectively). Furthermore, their expression levels, as assessed by MFI analysis, supported these findings: mD7 CAR T cells exhibited significantly higher expression of LAG-3 and PD-1 compared to the A5-derived constructs (Figure 5B). These combined results suggest that the mD7 CAR T cells exhibited a stronger activation profile associated with higher cytotoxicity and increased exhaustion, whereas the A5-derived CAR T cells—particularly hA5—displayed lower activation, lower cytotoxicity, and lower exhaustion levels under the same conditions.

### 3.6. CAR T Cell Cytokine Release

Cytokine secretion of the CAR T cells was assessed after 48 h of co-culture with PSMA-positive target cells. Levels of tumor necrosis factor (TNF), interleukin-2 (IL-2), interferon-gamma (IFNγ), granulocyte–macrophage colony-stimulating factor (GM-CSF), and Granzyme B (GrB) were quantified using cytometric bead assay (Figure 6A–E).

Untreated T cells secreted minimal cytokine levels, largely undetectable on the scales used. Among the CAR constructs, the D7-derived CAR T cells produced higher levels of several pro-inflammatory cytokines. Specifically, mD7 CAR T cells secreted the highest levels of IL-2 (1.2 ng/mL), IFNγ (5 ng/mL), and GM-CSF (6.8 ng/mL), followed by hD7 with moderately lower output. In contrast, A5-derived CAR T cells—particularly mA5—secreted the highest levels of Granzyme B (mA5, 99 ng/mL; hA5, 57 ng/mL), suggesting a distinct effector profile. Notably, secretion of Granzyme B showed substantial donor-dependent variability, ranging from 11 to 150 ng/mL for mA5. These findings suggest that, while D7-based CARs preferentially induce a pro-inflammatory cytokine profile, A5-based constructs favor granule-mediated cytotoxic responses.

## 4. Discussion

Improving CAR T cell persistence and reducing immunogenicity are central goals in enhancing therapeutic efficacy for both hematological and solid tumor targeting CAR T cells [3,4,5]. Humanization is a promising strategy to reduce immunogenicity and minimize rejection by the host immune system [16]. However, modifying parts of the scFv can lead to dramatic changes in CAR functionality, potentially compromising efficacy [24]. In this study, we demonstrate that our newly developed humanized PSMA-targeting CAR T cells retain strong, antigen-specific function in vitro, comparable to their murine counterparts.

### 4.1. Binding of Humanized CAR T Cells

Alterations in the binding properties of antibodies are frequent consequences of humanization procedures and may significantly affect the functional characteristics of CAR T cells, when these antibodies are incorporated as scFvs into CAR backbones [36,37]. In a recent study, a humanized scFv against CD19 showed an about 2000-fold reduced affinity compared to the murine parental scFv. Nevertheless, CAR T cells with this scFv showed robust proliferation without increased exhaustion and good antitumor activity as well as comparable in vivo antitumor activities as the CAR T cells with the murine scFv [38]. Moreover, a comparison of third-generation CAR T cells incorporating different humanized or fully human scFvs targeting carcinoembryonic antigen (CEA) revealed that CAR T cells equipped with a high-affinity scFv exhibited the highest levels of cytokine expression. However, these cells did not display increased cytotoxicity compared to CAR T cells expressing a scFv with moderate affinity. In a murine tumor model, CAR T cells with moderate affinity even outperformed their high-affinity counterparts [39]. CAR T cell cytotoxicity and cytokine release have previously been shown to be selectively activated depending on the range of CAR affinities [40]. Accordingly, modulation of CAR affinity can increase CAR T cell safety by balancing target recognition with CAR T cell-mediated on-target off-tumor side effects [41,42]. Given these observations we investigated how humanization affected the affinity of our hD7 and hA5 constructs.

In our study, humanization of the variable domains of D7 and A5 led to increased or decreased affinity in the hIgG1 format, respectively. We did, however, not observe significant functional differences in the CAR T cells with the humanized scFvs compared to the murine counterparts. Since affinity of the CAR T cells has to be judged in consideration of target antigen density, CAR expression levels, and the spatial relationship between the epitope and the CAR, affinity-dependent differences might become apparent during the in vivo characterization of the humanized CAR T cells [43].

### 4.2. Generation, Activation, and Tonic Signaling of Humanized CAR T Cells

Although the hD7 and A5 CAR constructs showed slightly lower transduction efficiencies and surface expression levels compared to the mD7 CAR, the expression remained sufficient to elicit robust antigen-specific activation and comparable cytotoxicity on prostate cancer cells. Optimization of scFv sequences or transduction protocols may further improve expression levels. Differences in CD25 expression and cytokine profiles across constructs suggest subtle differences in activation kinetics, potentially due to scFv affinity or tonic signaling [44]. These distinctions may influence downstream signaling, differentiation, and exhaustion profiles and will require in vivo validation.

The minor increase in CD25 expression observed in some CAR T cells following co-culturing with PSMA-negative targets may reflect low-level tonic signaling. However, the limited tonic signaling did not appear to have a negative impact on the specificity profile of the developed CAR T cells. No CAR T cells showed a significantly higher cytotoxicity compared to UT T cells when co-cultured with PSMA-negative target cells. Determining whether such signaling enhances functional persistence or contributes to premature exhaustion in the tumor microenvironment remains an important future objective [45].

### 4.3. Cytotoxicity of Humanized CAR T Cells

All CAR T cell constructs exhibited potent cytotoxicity against PSMA-positive target cells at high E:T ratios. The humanized CAR T cells retained the cytotoxic profiles of the murine derived CAR T cells with no differences between murine and humanized A5 or D7 derived CAR T cells emerging. Reduced killing at lower ratios revealed differences between constructs, with D7-based CAR T cells—particularly mD7 based—demonstrating superior performance. This may reflect either higher scFv affinity or expression levels. The enhanced lysis of PC3-PSMA cells, which express high PSMA levels (Appendix A) supports antigen density as a modulator of killing efficiency [46].

Interestingly, untransduced (UT) T cells also mediated detectable killing at high E:T ratios when co-cultured with PC3-PSMA cells, which were derived from PC3 cells exhibiting features of prostatic small-cell carcinoma and a near-triploid karyotype [47,48]. Although these tumor cells may display reduced or altered HLA class I expression, the HLA mismatch with donor-derived T cells used for CAR generation likely elicit alloreactive responses, particularly under conditions of high effector-to-target cell excess [49]. In addition, PC3-derived cells are known to upregulate stress ligands such as MICA, MICB, or ULBPs, which can activate endogenous T cell subsets, including CD8+ and γδ+ T cells, via the NKG2D axis [50,51,52]. Together, these mechanisms may account for the antigen-independent cytotoxicity observed in UT T cells.

Whether the moderate reduction in cytotoxicity seen with A5-based CARs is functionally advantageous remains to be determined. In vivo, reduced over-activation may preserve CAR T cell fitness and mitigate exhaustion. Conversely, stronger tonic signaling in D7 CAR T cell might be required to overcome suppression in the solid tumor microenvironment.

### 4.4. Differentiation, Inhibition, and Cytokine Secretion of Humanized CAR T Cells

All CAR T cell products displayed antigen-induced differentiation. The mD7 CAR T cells yielded the most differentiated phenotype, with a higher proportion of effector-like cells and fewer Tn, scm-like subsets. In contrast, A5-derived CAR T cells retained a more memory-like phenotype, which may be favorable for long-term persistence and recall responses in vivo. Expression of exhaustion markers mirrored this pattern. mD7 CAR T cells expressed higher levels of PD-1 and LAG-3, suggesting stronger activation or chronic signaling. This was accompanied by elevated secretion of IL-2, IFN-γ, and GM-CSF. Conversely, A5 CAR T cells—especially mA5-derived CAR T cells—secreted more Granzyme B, indicating a shift toward granule-mediated cytotoxicity.

Importantly, cytokine release syndrome (CRS), one of the most common toxicities of CAR T cell therapy, is driven by cascades of inflammatory cytokines, including IL-6, GM-CSF, TNF, and IFNγ [53,54,55]. With hA5 CAR T cells producing the lowest levels of TNF, IFNγ, and GM-CSF, they may present a reduced risk of initiating or amplifying CRS while still maintaining cytotoxic efficacy. This cytokine profile highlights a potential safety advantage of the hA5 construct, as lower pro-inflammatory output could mitigate systemic immune activation without compromising tumor-directed activity. These distinct functional signatures underscore how the scFv sequence directly shapes CAR behavior and support further investigation of hA5 CAR T cells as a safer alternative in preclinical CRS models [24,56].

### 4.5. Modeling of CAR Rejection: Challenges, Solutions and Clinical Relevance

Studying the immunogenicity of CAR constructs—particularly in the context of solid tumors—remains challenging. While xenograft models in immunodeficient mice are useful for assessing tumor clearance, they do not recapitulate key elements of immune rejection or the tumor microenvironment [57,58,59]. Syngeneic mouse models, though immunocompetent, are unsuitable for testing humanized CARs due to interspecies incompatibility. Humanized mouse models may offer a suitable compromise for investigating both immune tolerance and efficacy [27,57]. Beyond increasing persistence, humanization may facilitate the development of allogeneic or “off-the-shelf” CAR T cell therapies by reducing host-versus-graft responses [60].

### 4.6. “Off-the-Shelf” CAR T Cells and Targeting Metastatic Prostate Cancer

“Off-the-shelf” CAR T cells incorporate different genetic modifications to reduce their immunogenicity and off-target reactivity to enable an allogeneic transfer of CAR T cells [61]. Targeted integration of the CAR into the *T cell receptor alpha constant region (TRAC)* locus is one way to decrease the risk of graft-versus-host disease [62]. The physiological transcription regulation of a promoter-less CAR through integration into the *TRAC* locus has previously shown increased CAR T cell efficiency [63]. Knocking-out MHC class I via disruption of *B2M* and knocking-in non-classical HLA molecules like HLA-E can improve resistance of the transferred product to host-versus-graft reactions by inhibiting NK-mediated rejection [64,65,66].

In parallel, CAR NK cells are emerging as an attractive “off-the-shelf” alternative. Unlike T cells, NK cells are naturally less dependent on HLA compatibility, which facilitates allogeneic application with a favorable safety profile [67]. CAR NK cells not only exert CAR-mediated killing but also retain their intrinsic NK cytotoxicity, allowing them to eliminate target cells even in the case of antigen downregulation [68]. While their therapeutic use may be challenged by limited persistence and tumor infiltration in solid cancers [69], ongoing approaches to enhance CAR NK cell expansion and in vivo persistence, including cytokine support and genetic engineering, are critical to unlock their full potential [70,71]. Humanization may improve “off-the-shelf” CAR T and NK designs by further reducing the immunogenicity of an allogeneic CAR T/NK cell product. A major clinical benefit of “off-the-shelf” CAR immune cells is the significantly reduced time to treatment initiation, which is particularly advantageous for patients with advanced or systemic disease [72].

To treat systemic disease, a high persistence and efficacy of CAR T cells is necessary. In previous animal studies PSMA-targeting CAR T cells have shown their ability to eradicate focal prostate cancer tumors upon intratumoral application. However, in a model of systemic disease, a combinatorial treatment consisting of docetaxel chemotherapy and subsequent intravenous application of CAR T cells has only been able to slow tumor growth [25]. Insufficient persistence has been identified as one obstacle to successful CAR T cell therapy in prostate cancer [73]. Increasing the persistence of PSMA-targeting CAR T cells through humanization may improve therapeutic efficacy in the setting of metastatic prostate cancer, which currently remains incurable.

To summarize, we demonstrated that PSMA-targeting CAR T cells with humanized scFvs retain robust antigen-specific function in vitro. Humanization reduced murine sequence content without compromising cytotoxicity, activation, differentiation, or cytokine secretion. These results lay the foundation for further preclinical testing and potential translation into clinical applications targeting solid tumors, in general, and (metastatic) prostate cancer specifically.

## Figures and Tables

**Figure 1 cells-14-01333-f001:**
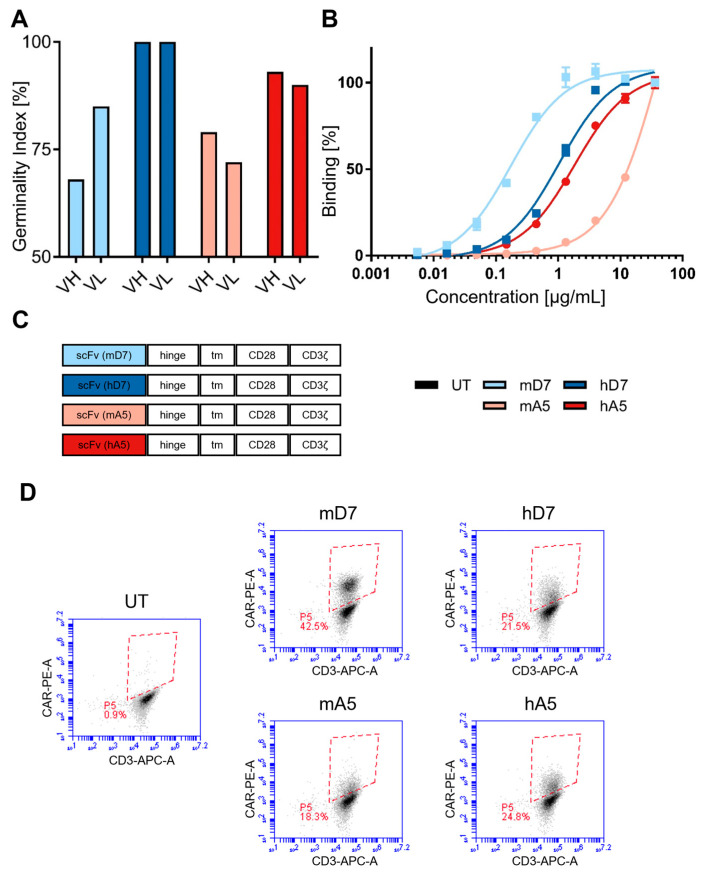
Humanization of CAR T cells: (**A**) Germinality indexes (GIs) of the variable heavy (VH) and light (VL) chains of the scFvs. (**B**) Binding of the VH/VL combinations in hIgG1 format to PSMA-positive C4-2 cells. Assessment is performed in technical triplicate, mA5 36 µg/mL in technical duplicate. (**C**) CAR structure with the different scFvs incorporated. All CARs are expressed under the elongation factor 1α short (EFS) promoter. The hinge region carries an additional modification to reduce Fc receptor interactions [30]. The transmembrane (tm) domain is derived from CD28. The CD28 costimulatory domain is deficient in LcK binding [31]. All CARs signal through CD3ζ. (**D**) Fraction of CAR-positive T cell levels after γ-retroviral transduction. A representative example is shown.

**Figure 2 cells-14-01333-f002:**
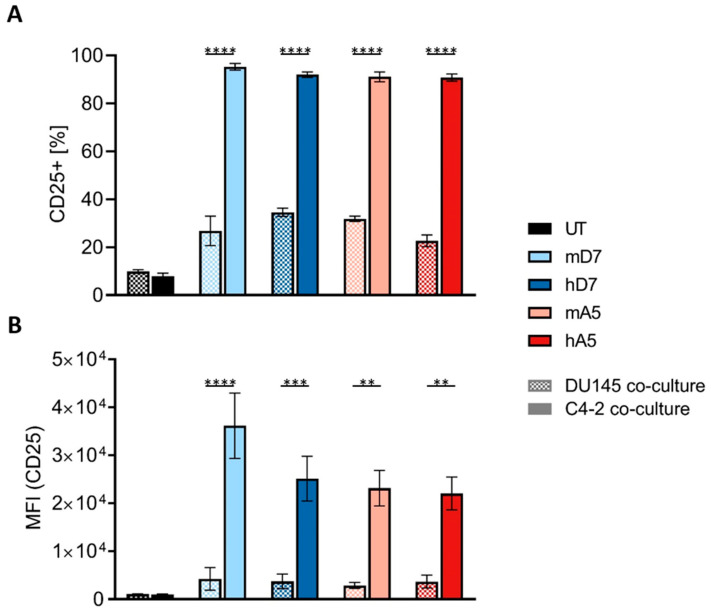
CAR T cell activation profiles: CAR T cells are co-cultured with either C4-2 (PSMA+) or DU145 (PSMA-) target cells. Activation is assessed by monitoring the expression of CD25. (**A**) Percentage of CD25-positive cells. (**B**) Mean fluorescent intensity (MFI) of CD25 expression. Patterned bars: Co-culture with DU145 cells. Solid bars: Co-culture with C4-2 cells. Abbreviations: UT—untreated T cells; mD7—murine D7 CAR; hD7—humanized D7 CAR; mA5—murine A5 CAR; and hA5—humanized A5 CAR. Illustrated are means +/− SEM of three biological replicates performed in technical triplicates. The statistical analysis is performed in GraphPad Prism 7 using a two-way ANOVA with Sidak’s multiple comparisons test; **; ***; and **** indicate *p* < 0.01, *p* < 0.001, *p* < 0.0001, respectively.

**Figure 3 cells-14-01333-f003:**
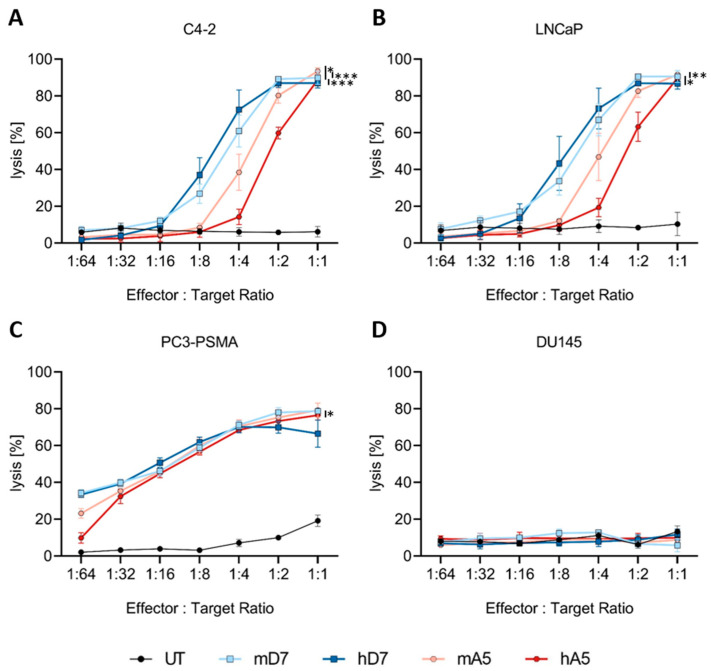
Cytotoxicity of the PSMA-targeted CAR T cells on prostate cancer cells. Untreated (UT) T cells and CAR T cells are co-cultured with different PSMA+ (**A**–**C**) and PSMA- (**D**) target cell lines. Depicted is the mean and standard error of the mean (SEM) of three biological replicates, each performed in technical triplicates. Comparison of mD7 vs. hD7 or mA5 vs. h is non-significant in all co-cultures. Significance in A: mD7 vs. hA5 *p* = 0.0008; hD7 vs. mA5 *p* = 0.0392; and hD7 vs. hA5 *p* = 0.0006. Significance in B: mD7 vs. hA5 *p* = 0.007; hD7 vs. hA5 *p* = 0.0104. Significance in C: mD7 vs. hA5 *p* = 0.0125. To reduce visual clutter no significance bars comparing UT with CAR T cell conditions are shown. In all PSMA+ co-cultures (C4-2, LNCaP, PC3-PSMA, all CAR T cell conditions mediated statistically significant higher lysis when compared to UT T cells. The statistical analysis was performed in GraphPad Prism 7 using a two-way ANOVA with Tukey’s multiple comparisons test. *; **; and *** indicate *p* < 0.05, *p* < 0.01, *p* < 0.001, respectively. Abbreviations: UT—untreated T cells; mD7—murine D7 CAR; hD7—humanized D7 CAR; mA5—murine A5 CAR; and hA5—humanized A5 CAR.

**Figure 4 cells-14-01333-f004:**
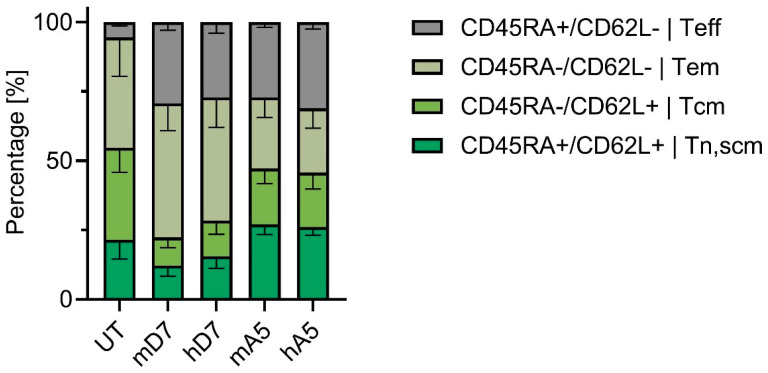
Phenotypes after exposure to cognate CAR antigen. Expressions of CD45RA and CD62L are used to assess the differentiation capacity of the CAR T cells upon antigen encounter. Abbreviations: Tn, scm—T_naïve/stem cell memory like_, Tcm—T_central memory like_, Tem—T_effector memory like_ and Teff—T_effector like_. UT—untreated T cells; mD7—murine D7 CAR; hD7—humanized D7 CAR; mA5—murine A5 CAR; and hA5—humanized A5 CAR. Depicted is the mean and standard error of the mean (SEM) of three biological replicates, each performed in technical triplicate.

**Figure 5 cells-14-01333-f005:**
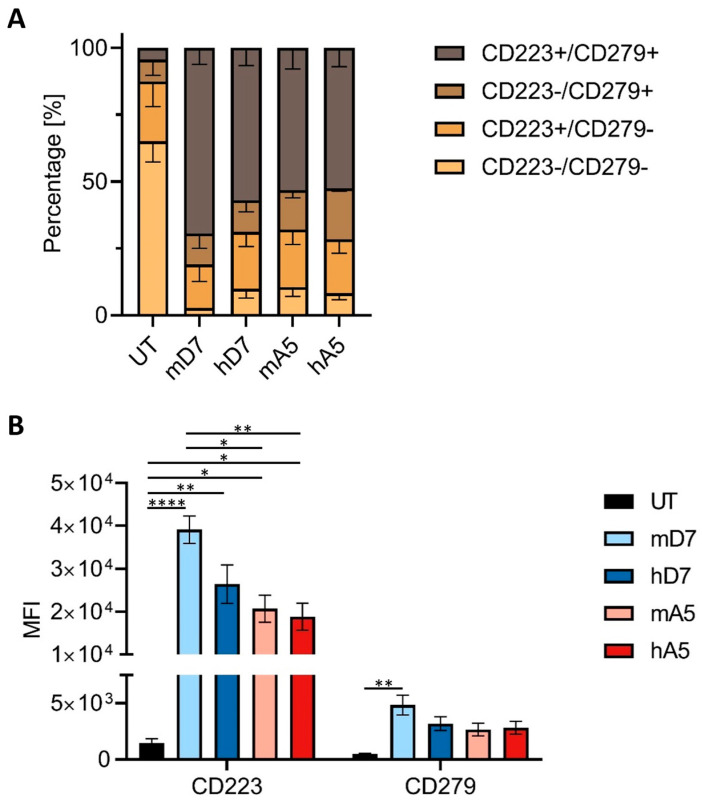
Expression of the inhibitory receptors PD-1 (CD279) and LAG-3 (CD223) after exposure to PSMA+ target cells: (**A**) Fraction of PD-1 and LAG-3 positive cells. (**B**) Mean fluorescent intensity of PD-1 and LAG-3 expression. Abbreviations: UT—untreated T cells; mD7—murine D7 CAR; hD7—humanized D7 CAR; mA5—murine A5 CAR; and hA5—humanized A5 CAR. (**B**) Illustrated are the means and standard error of the mean (SEM) of three biological replicates performed in technical triplicates. The statistical analysis is performed in GraphPad Prism 7 using one-way ANOVA; *; **; and **** indicate *p* < 0.05, *p* < 0.01, and *p* < 0.0001, respectively.

**Figure 6 cells-14-01333-f006:**
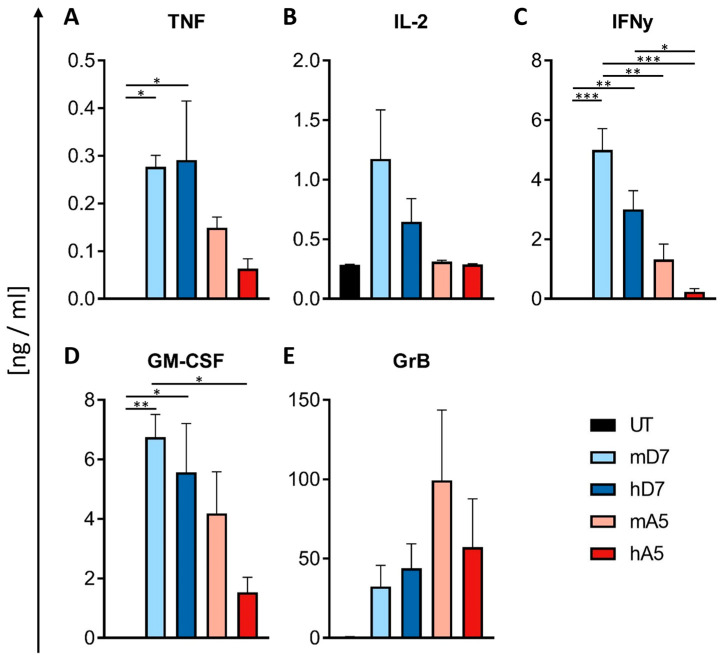
Cytokine release profile of CAR T cells after exposure to PSMA+ target cells. Levels of secreted (**A**) TNF, (**B**) IL-2, (**C**) IFNγ, (**D**) GM-CSF, and (**E**) Granzyme B (GrB) are assessed by cytometric bead assay. Abbreviations: UT—untreated T cells; mD7—murine D7 CAR; hD7—humanized D7 CAR; mA5—murine A5 CAR; hA5—humanized A5 CAR. (**B**) Means and standard error of the mean (SEM) of three biological replicates performed in technical triplicates. The statistical analysis is performed in GraphPad Prism 7 using one-way ANOVA; *; **; and *** indicate *p* < 0.05, *p* < 0.01, and *p* < 0.001, respectively.

## Data Availability

The datasets generated during and/or analyzed during the current study are available from the corresponding author upon reasonable request.

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
