# Peer review of "Engineering of Humanized PSMA-Directed CAR T Cells for Potent and Specific Elimination of Prostate Cancer Cells"

_cells, 2025, doi:10.3390/cells14171333_

Round 1
Reviewer 1 Report
Comments and Suggestions for Authors
The work presented is a good step towards improving CAR T cell therapy for prostate cancer. The in vitro study is logical and one of the humanized CAR construct (hA5) seems to be more useful in preventing cytokine storms, which is a major problem in CAR T cell therapy. An in vivo study using humanized mouse model would clearly strengthen the significance of this study before going into any clinical studies.
The in vitro studies are logical and well addressed. Should emphasize the advantage of hA5 CAR T cell construct with respect to cytokine storms in CAR T cell therapy.
In the Discussion, you should include “Off-the-Shelf” CAR -NK cells, along with the discussion of “Off-the-Shelf” CAR T cells.
Please correct the numbering of references in the text (it stars with reference 3, in introduction, instead of 1).
In Fig.3C, UT shows killing at high ET ratio, please explain.

Reviewer 2 Report
Comments and Suggestions for Authors
In this preclinical in vitro study, the authors characterized humanized CAR T cells targeting PSMA in cultured prostate cancer cell lines. They demonstrated that humanization improved the germinality index while preserving CAR expression on the cell surface. The humanized CAR T cells maintained antigen-specific function in vitro. Importantly, humanization did not compromise cytotoxicity, activation, differentiation, or cytokine secretion. The authors suggest that these findings may support future clinical applications targeting prostate cancer and solid tumors more broadly. The manuscript is illustrated with six figures and two supplementary figures and includes 58 references. It is logically structured and meticulously prepared. It can be accepted in its present form.
Author Response
We thank the reviewer for this very positive evaluation.